# High-Content Drug Discovery Targeting Molecular Bladder Cancer Subtypes

**DOI:** 10.3390/ijms231810605

**Published:** 2022-09-14

**Authors:** Sébastien Rinaldetti, Qiong Zhou, Joshua M. Abbott, Florus C. de Jong, Hector Esquer, James C. Costello, Dan Theodorescu, Daniel V. LaBarbera

**Affiliations:** 1The Department of Pharmaceutical Sciences, The Skaggs School of Pharmacy and Pharmaceutical Sciences, The University of Colorado Anschutz Medical Campus, Aurora, CO 80045, USA; 2The Department of Hematology-Oncology, Centre Hospitalier de Luxembourg, 1210 Luxembourg, Luxembourg; 3The Department of Urology, Erasmus MC Cancer Institute, 3015 GD Rotterdam, The Netherlands; 4Department of Pharmacology, University of Colorado Anschutz Medical Campus, Aurora, CO 80045, USA; 5The University of Colorado Cancer Center, University of Colorado Anschutz Medical Campus, Aurora, CO 80045, USA; 6Cedars-Sinai Samuel Oschin Comprehensive Cancer Institute, Los Angeles, CA 90048, USA; 7The Center for Drug Discovery, University of Colorado Anschutz Medical Campus, Aurora, CO 80045, USA

**Keywords:** bladder cancer, drug discovery, molecular subtype

## Abstract

Molecular subtypes of muscle-invasive bladder cancer (MIBC) display differential survival and drug sensitivities in clinical trials. To date, they have not been used as a paradigm for phenotypic drug discovery. This study aimed to discover novel subtype-stratified therapy approaches based on high-content screening (HCS) drug discovery. Transcriptome expression data of CCLE and BLA-40 cell lines were used for molecular subtype assignment in basal, luminal, and mesenchymal-like cell lines. Two independent HCSs, using focused compound libraries, were conducted to identify subtype-specific drug leads. We correlated lead drug sensitivity data with functional genomics, regulon analysis, and in-vitro drug response-based enrichment analysis. The basal MIBC subtype displayed sensitivity to HDAC and CHK inhibitors, while the luminal subtype was sensitive to MDM2 inhibitors. The mesenchymal-like cell lines were exclusively sensitive to the ITGAV inhibitor SB273005. The role of integrins within this mesenchymal-like MIBC subtype was confirmed via its regulon activity and gene essentiality based on CRISPR–Cas9 knock-out data. Patients with high ITGAV expression showed a significant decrease in the median overall survival. Phenotypic high-content drug screens based on bladder cancer cell lines provide rationales for novel stratified therapeutic approaches as a framework for further prospective validation in clinical trials.

## 1. Introduction

Over the past 40 years, the survival rate associated with bladder cancer (BLCA) has not significantly changed [1]. Methotrexate, vinblastine, adriamycin, and cisplatin (MVAC) and cisplatin/gemcitabine have been the standard of care in muscle-invasive bladder cancer (MIBC) for decades. Targeted therapies have been limited to recent breakthroughs, including FGFR and immune-checkpoint inhibitors [2,3]. We postulate that the stagnation of BLCA-targeted therapies has been caused, in part, by insufficient molecular stratification of study patients and a lack of valid companion biomarkers. Comprehensive transcriptomic studies recently identified distinct molecular subtypes that can be condensed to basal, luminal, mesenchymal/stroma-like, and neuroendocrine gene signatures [4,5]. Stratifying patients by these molecular subtypes improved survival prediction and displayed differential drug sensitivities in clinical trials [6,7,8]. Conversely, the potential of using molecular subtypes as a paradigm for phenotypic drug discovery has not yet been exploited. Thus, the overarching goal of this study was the identification of subtype-specific BLCA lead drugs.

BLCA cell lines play a pivotal role in drug discovery for several reasons. First, there exists a high number of well-characterized cell lines that reflect crucial genetic aberrations also found in MIBC patients (e.g., TERT promotor mutations, FGFR3-TACC3 fusion genes) [9]. Second, cell line-based drug sensitivity gene signatures can be translated into clinical predictive models [10,11]. Third, the ongoing development of comprehensive in silico databases, established on these cell lines, may help accelerate the pace of novel discoveries. The cancer dependency map database synergizes data from the Project Achilles (genome-wide loss-of-function screens), Cancer Cell Line Encyclopedia (CCLE, multiomic profiling) and the PRISM Repurposing Screen (single or pooled compound screens) [12].

With the increased molecular genetic understanding of BLCA, novel-targeted therapies are emerging. Recent studies have shown promising preclinical data for EZH2 and BET inhibitors [13,14]. Ongoing phase 1 and 2 trials are testing FGFR inhibitors, PARP inhibitors, and mTOR inhibitors for their clinical activity in MIBC patients [15]. Promising investigational new drugs, such as the HDAC inhibitor, Mocetinostat, and the PARP inhibitor, Rucaparib, have failed to show efficacy in an unselected bladder cancer patient cohort [16,17]. The mesenchymal/stromal-like molecular subtype is of special interest since numerous studies have demonstrated its resistance to chemotherapy [4,5,18,19]. Several taxonomies for this phenotype are in use, depending on the classifier: mesenchymal-like, p53-like, stroma-rich, and luminal-infiltrated [4,5,18,20]. This BLCA type is characterized by epithelial–mesenchymal transition (EMT) markers, wild-type TP53, B-cell infiltration, smooth muscle, and myofibroblast infiltration [4,5].

To identify subtype-specific lead drugs and gain insight into the molecular subtype biology, we utilized a high-content screening (HCS) phenotypic drug discovery approach, using two focused drug libraries screened independently. We classified existing BLCA cell lines into the major molecular muscle-invasive BLCA subtype taxonomies. By combining drug sensitivity data with functional genomics, regulon, and drug set enrichment analysis, we were able to identify and validate both established drugs and novel lead drugs effective against subtype-specific BLCA cell lines.

## 2. Results

### 2.1. Molecular Subtyping of Bladder Cancer Cell Lines

A total of 36 BLCA CCLE cell lines and 36 cell lines of the BLA-40 panel were classified into the major known molecular subtype taxonomies based on the Lund, MDA, TCGA, and Consensus classifiers [4,5,18,20]. The CCLE and the BLA-40 cell line libraries have 16 cell lines in common. The assignments to their predominant molecular subtypes were identical for 14 of those 16 cell lines, except for UMUC1 and UMUC14 (Figure 1A and Figure 2B). The basal and luminal subtypes presented a distinct overlap throughout the different taxonomies. To validate those subtypes, we identified the most enriched gene signatures using gene set enrichment analysis (Appendix A). As expected, the basal subtype was highly enriched with curated basal signatures. The luminal subtype also showed a distinct enrichment with luminal gene sets and was strongly associated with peroxisomal signatures. In fact, PPARG is known to be a major player in luminal phenotype determination in BLCA patients [4,21].

Some cell lines were mainly characterized by mesenchymal and smooth muscle gene signatures (Appendix A). The Lund classification identified most of those cell lines as mesenchymal. In the same context, the MDA classifier identified those cells as “p53-like” or “basal”. This specific overlap of the Lund and MDA classifiers has been shown before [5,20]. In fact, the p53-like subtype covers stem cell-like gene signatures, while the basal subtype also presents mesenchymal gene expression [20]. However, the TCGA and consensus classifier use stromal infiltration markers of fibroblasts and leukocytes for their mesenchymal-like subtypes (e.g., “luminal infiltrated” of the TCGA and “stroma-rich” of the Consensus classifier), which are not present in pure cell lines. Thus, an erroneous classification of these cell lines to the neuronal subtype is probable when considering that only a very small proportion of BLCA patients display a neuronal subtype. Finally, we designated these cell lines as “mesenchymal-like” because the number of concordant classifiers was always less than three, and the enriched gene signatures confirmed an intrinsic mesenchymal-like phenotype.

### 2.2. CCLE BLCA Cell Lines: Subtype-Specific Drug Sensitivities

A total of 1448 drugs of the CCLE HCS were grouped by their drug targets defining the “drug sets” (Appendix A). Based on enrichment analysis, we determined the false discovery rate for each drug set to investigate their subtype-specific impact on cell survival. The basal cell lines displayed a distinct sensitivity for inhibitors of cell cycle proteins, such as CHK1/2, PLK, and thymidylate synthase, and to a lesser extent for RAF and AKT (Figure 1B). The CHK1 inhibitor Rabusertib (LY2603618) showed the highest subtype-specific sensitivity to basal BLCA cells (Figure 1C). Docetaxel, another lead hit in basal cell lines, is a microtubule-targeting drug primarily used to treat various squamous carcinoma but also effective in bladder cancer resistant to standard of care therapy (Figure 1C) [22]. Squamous carcinoma and urothelial carcinoma are the two most frequent histological BLCA manifestations. Squamous BLCA patients are always classified into the basal molecular subtype and show poor survival [4]. To date, docetaxel has not been tested in selected basal or squamous BLCA patients only.

The most selective growth inhibition of luminal BLCA cells was mediated via the inhibition of MDM2, a negative regulator of p53 (Figure 1B) [23]. Accordingly, the MDM2 inhibitor, Idasanutlin, displayed the highest subtype-specific growth inhibition in luminal cell lines (Figure 1C). The luminal cell lines were also sensitive to mTOR inhibitors (Figure 1B), especially Ridaforolimus (also known as Deferolimus) and Temsirolimus. Patients with luminal BLCA have a high frequency of FGFR mutational changes and increased FGFR transcript levels [4,18,24]. In line with this, our analysis shows that luminal cells were sensitive to the FGFR1/3 inhibitor PD173074 (Figure 1C).

The AUC value of the mesenchymal-like cell lines remained high even for the most selective agents, demonstrating a certain degree of resistance against most drugs (Figure 1C). The only anticancer agents that showed a significant subtype-specific inhibitory effect were inhibitors of the Rho-associated kinase (ROCK), which is a mesenchymal biomarker known to play a major role in cell adhesion and cytoskeleton organization (Figure 1B) [25].

### 2.3. Focused In-House HCS against BLCA Subtypes

The findings based on the CCLE HCS demonstrate that BLCA cell lines represent a valid platform for subtype-specific drug discovery. Consequently, in contrast to the PCR-based CCLE HCS, we performed a multiplexed (Hoechst 33342 [ThermoFisher, Waltham, MA, USA] and Celltox™ green [Promega, Madison, WI, USA]) high-content screen based on a focused Selleck Chem drug library including 616 novel clinical compounds. For this HCS, we used preselected representative cell lines belonging to the BLA-40 library (luminal: RT4, JON, UMUC9 basal: PSI, VMCUB1, SCABER, mesenchymal-like: UMUC3, 253Jbv, J82). The assessment of the drug response was based on the percentage of viable cells after drug exposure. We clustered the selected cells based on their drug response to 616 clinical or pre-clinical drugs of the Selleck Chem library (Appendix A). We found that cell lines that were assigned to an identical subtype showed similar drug responses. This further supports that these are distinct molecular subtypes with inherent subtype-specific molecular features. Finally, we chose a total of five lead drugs presenting the highest subtype-specific growth inhibition for a given subtype for further validation (Appendix A).

Similar to the CCLE HCS, the basal cell lines were sensitive to cell cycle inhibitors (Figure 2B). Furthermore, they showed a distinct response to histone deacetylase (HDAC) inhibitors. Among the top hits, two epigenetic modulators displayed distinct subtype-specific inhibition: Citarinostat, an HDAC inhibitor, and I-BET-762, a bromodomain and extra-terminal motif (BET) inhibitor (Figure 2C). As shown by the dose–response curves, the basal cell lines exhibited an enhanced sensitivity for Citarinostat (*p* = 0.01, Figure 3). The IC_50_ values differed significantly between the basal and mesenchymal-like subtypes, with a less pronounced effect against the luminal subtype. The basal cell lines also displayed sensitivity to mTOR inhibitors in drug set enrichment analyses, but they did not represent lead hits (Figure 2C and Appendix A). Since the CCLE drug screen showed sensitivity to mTOR inhibitors in luminal cells, a subtype-specific action of mTOR inhibitors cannot be confirmed. We observed significant inhibition of luminal BLCA cell growth with PARP inhibitors, estrogen receptor modulators, and neuromodulators (Figure 2B). However, the enrichment analyses yielded relatively high false discovery rates. Among the single compound lead hits, Umbralisib, a PI3K inhibitor, and MK-886, a lipoxygenase inhibitor, showed the most significant growth inhibition of luminal cells (Figure 2C and Appendix A). However, this subtype-specific growth inhibition could not be reproduced upon validation with dose–response curves (Figure 3).

The mesenchymal-like cell lines displayed subtype-specific sensitivity to Aurora kinase (AURK) inhibitors and other cell cycle inhibitors (Figure 2B), but as single agents, they did not appear in the top hits (Figure 2C and Appendix A). In accordance with the CCLE findings, cytoskeletal inhibitors presented high specificity in the growth inhibition of mesenchymal-like cell lines (Figure 2B). In particular, the integrin inhibitor SB273005 displayed outstanding and exclusive inhibition of mesenchymal-like cells (Figure 3 and Appendix A). SB273005 is a potent integrin inhibitor with K_i_ of 1.2 nM and 0.3 nM for the αvβ3 receptor and αvβ5 receptor, respectively [26]. Within the CCLE drug screen, the integrin and ROCK inhibitor, Simvastatin, also displayed specific action against the mesenchymal-like cell lines (Appendix A, *p* = 0.05).

### 2.4. Drug Target Validation by Regulon Analysis and Functional Genomics

To evaluate the inherent molecular biology of each subtype, we performed regulon analysis via VIPER, based on the RNA-seq data from the CCLE cell lines. VIPER allows a subtype-specific identification of relevant active and inactive proteins. In line with the sensitivity of basal urothelial carcinoma to HDAC inhibitors, we identified the arginine–glutamic acid dipeptide repeats gene (RERE) as a leading regulon active in basal BLCA cell lines (Figure 1D, *p* < 0.001). RERE is known for its role in embryogenesis and histone deacetylation [27,28]. A potential role in basal phenotype determination is not yet known. In addition, we found highly suppressed RALA protein activity in basal cells (Figure 1D, *p* < 0.001). The Ras-related GTPases (Ral-A and Ral-B) are oncogenes known to be activated via integrin-mediated downstream signaling [29,30,31]. The suppressed RALA protein activity may contribute to the complete resistance of basal cell lines toward the integrin inhibitor SB273005 (Figure 4).

The sensitivity of mesenchymal-like cell lines to integrin inhibitors is reflected by a strong activation of the integrin regulons ITGB4 and ITGB6 (*p* = 0.0016 and *p* = 0.0005 respectively, Appendix A). Mechanisms responsible for cellular motility and movement are, among others, based on the well-known interaction of integrins with ROCK1/2 [32,33,34]. Thus, the selective growth inhibition in mesenchymal-like CCLE cells by ROCK inhibitors such as Simvastatin is biologically plausible (Figure 1B, *p* = 0.023).

To confirm the role of integrin inhibitors in mesenchymal-like cell lines, we took a closer look at the genetic essentiality of integrins in the context of subtype-specific cell survival. We assessed the cell line-specific dependency scores of the different integrin gene variants via the genome-scale CRISPR–Cas9 knock-out data from the Project Achilles based on the CCLE cell lines. The CRISPR–Cas9-based knock-out of the ITGAV gene, coding for the α_v_ integrin subunit, showed essentiality in 46% of the mesenchymal-like cell lines, whereas only 11% of the basal cell lines and 0% of the luminal subtype were dependent upon ITGAV for cellular survival (Figure 4). A similar tendency was observed for the *ITGB5* gene coding for the β_5_ subunit of the integrin receptor (Figure 4).

### 2.5. TCGA Bladder Cancer Cohort: Clinical Impact of ITGAV

Since the α_v_β_3_ and α_v_β_5_ integrin inhibitor, SB273005, showed the most promising subtype-specific effect by targeting exclusively the mesenchymal-like cell lines, we investigated the impact of *ITGAV* on BLCA patient survival. The clinical impact of *ITGAV* expression was assessed by the TCGA cohort, which included 386 patients with muscle-invasive BLCA. Integrins are known to be EMT regulators [25]. In order to validate this relationship, we correlated the *ITGAV* expression with transcript levels of E-cadherin (*CDH1*) and vimentin (*VIM*), well-characterized biomarkers of the epithelial and mesenchymal phenotype respectively [35]. While *ITGAV* showed a strong correlation with the mesenchymal biomarker *VIM* (*p* < 0.001, R = 0.41, Figure 5A), there was no apparent correlation with the epithelial marker *CDH1* (*p* = 0.26, R = 0.06, Figure 5A). We further analyzed the differential *ITGAV* expression between the molecular MIBC subtypes. As expected, we found significant differential expressions of *ITGAV* between subtypes (Figure 5B). Patients with “basal” and “stroma-rich” (“stroma-rich” Consensus classifier = “luminal infiltrated” TCGA classifier, ref. [5]) presented the highest *ITGAV* expression in comparison to the other subtypes. The regulon analyses and CRISPR–Cas9 knock-out data mentioned above demonstrate the subordinated role of integrins in basal BLCA despite its high expression pattern in basal patients. Finally, we found that patients with high *ITGAV* expression had a median overall survival of only 26 months, which is in stark contrast to 59 months for patients with low *ITGAV* expression (Figure 5C, *p* = 0.036).

## 3. Discussion

In this study, we performed, for the first time, a systematic clustering of bladder cancer cell lines based on the major subtype taxonomies. We observed that a high number of well-characterized cell lines provide a strong tool for subtype-specific drug discovery in bladder cancer. Our phenotypic HCS drug discovery approach identified subtype-specific drug sensitivities that were consistent with clinical trials, as well as in vivo studies, and facilitated the discovery of novel subtype-specific lead drugs (Figure 6). Conversely, subtype-specific drugs allow for deeper insights into the inherent molecular pathophysiology of bladder cancer.

Patients with stromal or mesenchymal-like BLCA are known to be chemotherapy-resistant, and the landscape of actionable targets is limited [4,5,18,19]. Our multiplexed HCS approach identified integrins as potential targets against the mesenchymal-like BLCA cell lines. The integrin inhibitor SB273005 showed distinct growth inhibition of mesenchymal-like cell lines compared to basal and luminal cells. We validated this finding by regulon analysis and functional genomics. Of note, RALA, a major integrin downstream target, is deactivated in basal cell lines. Accordingly, RALA is known to be deactivated in squamous cancers [36]. Thus, the suppressed RALA protein activity may contribute to the inherent resistance of basal cell lines toward the integrin inhibitor SB273005 (Figure 3). Interestingly, mesenchymal-like CCLE cells were sensitive to simvastatin, which inhibits both integrin and ROCK. Simvastatin is also broadly used as a 3-hydroxy-3-methyl glutaryl (HMG)-CoA reductase (HMGR) inhibitor, which prevents cholesterol biosynthesis while lowering lipids [37]. Recent studies have discovered that the mesenchymal phenotype can be disrupted by targeting its lipid metabolism [35,38]. Consistent with our results, in a subtype independent study, Van der Horst et al. demonstrated that *ITGAV* knock-down had a greater impact on UMUC3 cell migration than on the RT4 cells [39]. In addition, based on a UMUC3 cell line-derived xenograft model, mice with *ITGAV* knock-down presented a markedly lower number of metastases and bone marrow infiltration than wild-type tumor-bearing mice [39]. Clinical validation of the TCGA cohort showed a distinct increase in the median overall survival in patients with low *ITGAV* expression (Figure 5). Numerous phase 1 and 2 trials are currently testing small molecules and monoclonal antibodies inhibiting integrin in advanced solid cancers [40]. A recent study demonstrated that integrins regulated PDL1 expression, and their depletion resulted in immunologically hot tumors [41]. When combined with immune checkpoint inhibitors, integrin depletion led to a durable response in vivo [41]. Taken together, integrin inhibitors in combination with immune checkpoint inhibitors are an attractive therapeutic option for patients with mesenchymal-like BLCA and deserve further prospective validation in clinical trials (Figure 6).

Our results also show that basal cell lines are sensitive to HDAC inhibitors. Accordingly, histone deacetylase was previously identified as a potential drug target in basal BLCA [42]. Within the CCLE drug screen, CHK1 inhibitors presented promising effects against basal cell lines. This result is supported by a recent study that showed CHK1 inhibitors being effective against basal-like breast cancer [43]. Interestingly, HDAC1/2 are known to sustain the phosphorylation and thus the activity of the checkpoint kinases CHK1/2, providing a link between these targets in BLCA [44]. MTOR inhibitors are effective in the growth inhibition of BLCA cell lines in general, but no distinct subtype-specific inhibitory effect was observed in this study [45,46], which is in line with other subtype-specific studies with mTOR inhibitors [47].

Concerning the luminal cell lines, Umbralisib (PI3K inhibitor) and MK-886 (lipoxygenase inhibitor) displayed no luminal-specific dose-dependent effect despite both pathways being known to be altered in BLCA [4,48]. The FGFR inhibitor PD-173074 proved to be a lead compound with efficacy against luminal BLCA cell lines in the CCLE HCS. In line with this, previous studies have identified FGFR alterations mainly in luminal bladder cancer [18,24,49]. Currently, the FGFR inhibitor Erdafinib is FDA-approved for advanced or metastatic BLCA patients with FGFR3 or FGFR2 mutations who have progressed in platinum-containing chemotherapy [50]. Luminal CCLE cells also displayed a distinct sensitivity toward MDM2 inhibitors. Likewise, the TCGA MIBC study showed major MDM2 pathway activity mainly in luminal muscle-invasive BCLA patients [4]. Thus, MDM2 inhibitors may prove to be effective anticancer agents for luminal BLCA patients.

Finally, we demonstrated a novel approach using enrichment analysis based on in-vitro drug response data. To date, drug set enrichment analyses were mainly used for the assessment of drug-induced gene expression signatures [51]. High-content drug screens generate large in vitro response data, facilitating enrichment analyses of focused target-based drug sets via the GSEA ranking algorithm. Similar to gene signatures, the enrichment of drug sets provides functional information. In contrast to single-agent analyses, the enrichment analyses link the common cellular target to the subtype-specific action of a drug class or drug set. This method can be adopted for phenotypic and target-based HCS drug discovery.

In conclusion, we demonstrated that phenotypic HCS drug discovery targeting BLCA subtypes can identify novel subtype-specific treatment options and reproduce known BLCA drug sensitivities (Figure 6). In particular, we identified integrins as a likely driver of the mesenchymal-like subtype in BLCA, with integrin inhibitor SB273005 displaying an exclusive inhibition of cell growth in mesenchymal-like BLCA cell lines. Subtype-directed HCS is a viable approach for the discovery of subtype-specific BLCA inhibitors, with the potential of accelerating the translational pace from bench to bedside. The results herein provide rationales for novel stratified precision medicine approaches as a framework for prospective validation in clinical trials.

## 4. Methods and Materials

### 4.1. Transcriptome Expression Analyses and Molecular Subtyping

Whole transcriptome RNA-seq data of 36 CCLE bladder cancer cell lines were obtained via DepMap Portal (DepMap Public 18Q4, accessed on 20 December 2018) [12]. Transcriptome expression analysis of the BLA-40 cell lines, a bladder cancer cell line cohort characterized independently of the CCLE, was performed by Affymetrix U133A arrays as described before [10]. All data were normalized and log2-transformed. In order to avoid redundancy in the subtype clustering and enrichment analyses, we omitted UMUC3e, 253jp, and 253jlaval from the BLA40 cell line cohort. Since UMUC2 is known to be a T24 contaminant, this cell line was also not included. Finally, we used a total of 36 cell lines from the BLA-40 dataset.

Bladder cancer cell lines were classified into the major molecular subtype taxonomies (MDA, Lund, TCGA, and Consensus classifiers) using the R-packages “BLCAsubtyping” and “ConsensusClassifier” available on GitHub [4,5,18,20]. In a second step, we assigned each cell line to its predominant molecular subtype. For this assignment, concordance of at least 3 of the 4 classifiers was required. Some subtypes within the classifiers were considered to be equivalent (e.g., the genomically unstable subtype based on the Lund classifier corresponds to the TCGA or MDA luminal subtype) [5,20]. This method showed high concordance between the CCLE and BL-40 cell lines and aligned with clinically validated MIBC subtypes.

### 4.2. Drug Set and Gene Set Enrichment Analyses

Drug set enrichment analysis was performed using the gene set enrichment analysis (GSEA) software 4.1.0 (Broad Institute, Cambridge, MA, USA) [52]. All drugs were assigned to their respective drug targets in the form of drug sets (Appendix A). These drug sets were adapted for the GSEA-based GMX format. The drug response data included AUC values based on the drug dilution curves of the CCLE PRISM Repurposing dataset (accessed on 20.12 2018, referred to here as “CCLE HCS”) or the percentage of viable cells based on the in-house focused high-content drug screen (referred to here as “focused in-house HCS”). The drug response data were adapted for the GCT format. The cell line phenotypes were defined as “basal”, “luminal”, and “mesenchymal-like”. Cell lines that were sensitive to pathway-specific drug sets show lower AUC (CCLE HCS) or a low number of viable cells (focused in-house HCS). Thus, we used GSEA software to screen for significantly negatively enriched drug sets. 

Next, gene set enrichment analysis was performed with R package GSVA 1.44.2 [53], which calculated the single sample normalized gene enrichment scores. We included gene signatures from MSigDB Hallmarks 7.2 (Broad Institute, MA, USA) and others [54]. The signatures were selected based on differential findings between the subtypes and visualized with heatmaps via R 4.2.1.

### 4.3. Regulon Analysis

Virtual inference of protein activity by enriched regulon analysis (Version 1.30.0) [55] was conducted to assess subtype-specific master regulator activities. The VIPER algorithm was run on subtype-specific differential expression analysis based on RNA-seq data of the CCLE cell line cohort. The context-specific regulatory networks (“aracne.networks”) used to infer the master regulator activities were downloaded from Bioconductor. In this study, we used the bladder cancer-specific “regulonblca” interactome as the network [56]. The *p*-values and normalized enrichment scores give information about the inferred protein activity.

### 4.4. Functional Genomics

We assessed candidate genes for their role in bladder cancer cell viability based on the genome-scale CRISPR–Cas9 knock-out data from the Project Achilles. Dependency scores were calculated using the CERES algorithm. Candidate genes with a CERES score of <−1 were identified as essential for cell survival (DepMap Public 18Q4, accessed on 20 December 2018) [57].

### 4.5. HCS Drug Discovery

The drug response of 36 BLCA CCLE cell lines to 1448 compounds was assessed by the PRISM Repurposing 19Q4 drug screen (CCLE HCS) [58]. The drug response curves consisted of an 8-step, 4-fold dilution, starting from 10 uM. The assay consisted of an mRNA-based read-out of genetically barcoded cell lines. The drug sensitivity was calculated using the area under the dose–response curve, with lower AUC values indicating increased sensitivity to the treatment.

For the focused in-house HCS, we used 616 novel clinical and pre-clinical inhibitors of the Selleck Chem Clinical Compound library. All compounds were diluted in DMSO. For each BLCA subtype, we screened 3 representative cell lines from the BLA-40 cell line library (luminal: RT4, JON, UMUC9; basal: PSI, VMCUB1, SCABER; mesenchymal-like: UMUC3, 253Jbv, J82). The cells were incubated in 384-well black flat-bottom cell culture microplates (Greiner Bio-one, Kremsmünster, Austria). The cells and reagents were plated with the automated Janus Liquid Handler Workstation (PerkinElmer, Waltham, MA, USA). Multiplexed imaging was performed in phenol red-free medium containing L-glutamine and supplemented with 10% FBS. The cells were plated on day 0, with 1000 cells per well. The drug library and CellTox™ Green Express (0.5X) reagent (Promega, Madison, WI, USA) were added on day 1 in singlets, with a final concentration of 10 µM and 0.5% DMSO. Lysis buffer was used as a positive control. After 72 h of incubation, the cells were stained with Hoechst 33342 (ThermoFisher, Waltham, MA, USA) for imaging and cell counting by the Opera Phenix HCS system (PerkinElmer, Waltham, MA, USA). The remaining cells with high Celltox™ green (Promega, Madison, WI, USA) signal were considered dead. The assay quality was assessed by the Z-prime value (Appendix A). A Z-prime value of >0.5 was considered acceptable. Images were taken with a 5× air objective. All cell lines were confirmed to be mycoplasma-free (MycoAlert, Lonza, Basel, Switzerland) and cultivated at 37 °C and 5% CO_2_. The cells were used in experiments within 6 weeks of thawing. The drug response curves (8-step log2 dilution, initial concentration = 40 µM) were set in quadruplets for the hit validation of five drugs in nine selected cell lines.

### 4.6. Clinical Data and Statistical Analysis

The RNA-seq data from 411 muscle-invasive BLCA (MIBC) patients of the TCGA cohort were evaluated for in silico validation. Patients with incomplete clinical and molecular annotation were excluded (*n* = 25), resulting in a cohort of 386 available MIBC patients [4,5,59]. The gene expression data were normalized and log2-transformed. The *ITGAV* cutoff for risk stratification based on overall survival was calculated by applying Youden’s index in conjunction with ROC analysis [60,61]. Overall survival was assessed by the Kaplan–Meier method and compared using the log-rank test. Differential gene or gene signature expression between the MIBC molecular subtypes was analyzed by the Kruskal–Wallis test. Drugs were ranked by their differential mean cell viability (in-house screen) or AUC (CCLE screen) between subtypes and tested for significance by unpaired two-tailed Student’s t-test with Welch correction for unequal variances. *p*-value asterisks criteria were adapted to the New England Journal of Medicine (NEJM) guidelines (<0.033 (*), <0.002 (**), <0.001 (***)). Statistical analyses were performed using RStudio (Boston, MA, USA), GraphPad Prism 7.03 (La Jolla, CA, USA), SPSS 25 (IBM, Chicago, IL, USA), and JMP 10 (SAS, Carry, NC, USA). 

## Figures and Tables

**Figure 1 ijms-23-10605-f001:**
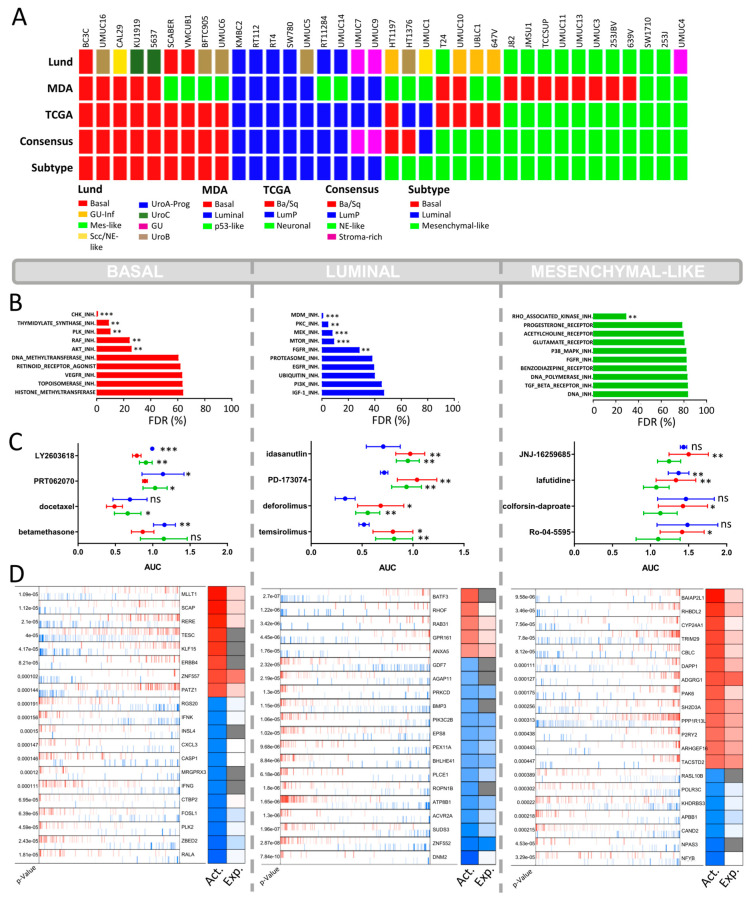
**CCLE cell lines high-content drug screen.** (**A**) Subclassification of the CCLE cell lines (*n* = 36) into the main known molecular BLCA subtype taxonomies: MDA, LUND, TCGA, and Consensus subtypes. The known subtypes converge into 3 main overlapping subtypes (“Subtype”): basal, luminal, and mesenchymal-like. Scc/NE-like = squamous cell carcinoma/neuroendocrine-like, GU-inf = genomical unstable infiltrated, UroA-Prog = Urobasal A progression, Mes-like = mesenchymal-like, LumP = luminal papillary, Ba/Sq = basal/squamous. (**B**) Drug set enrichment analysis based on the AUC values of the PRISM Repurposing dataset. A total of 1448 drugs were classified into 77 specific drug sets (e.g., MTOR inhibitors. PI3K inhibitors). FDR = false discovery rate. (*p* < 0.002: **, *p* < 0.001: ***) (**C**) Top 4 hits with the highest differential growth inhibition between the basal, luminal, and mesenchymal-like BLCA subtypes. NS = non-significant (*p* < 0.033: *, *p* < 0.002: **, *p* < 0.001: ***) (**D**) Virtual inference of protein-activity by enriched regulon analysis (VIPER) displaying differential regulon activity (first column: red = activated; blue = inactivated) and regulon expression (second column: red = high expression; blue = low expression; gray = not applicable) between subtypes. Representation of the top 20 most significantly activated or inactivated regulons.

**Figure 2 ijms-23-10605-f002:**
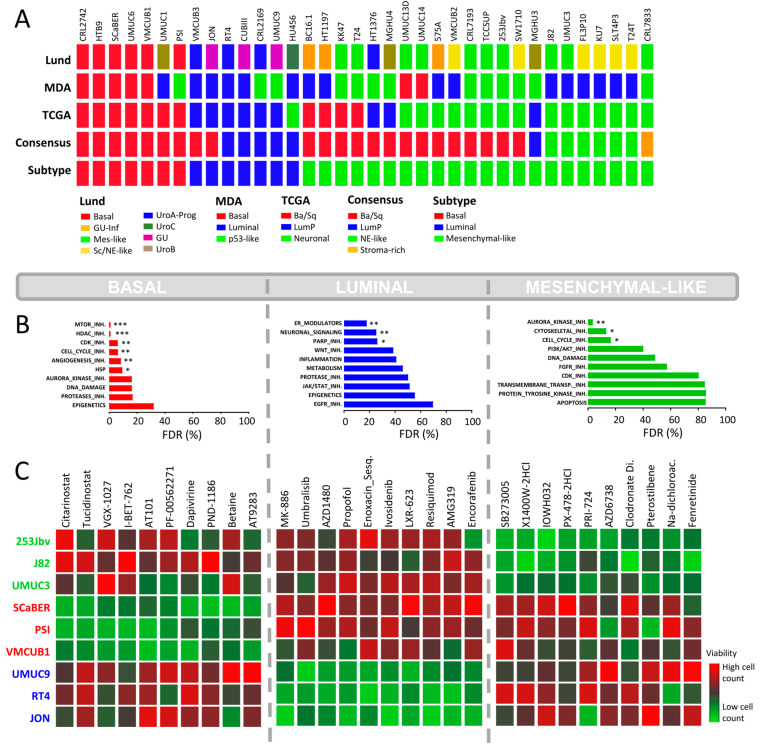
**In-house BLA-40 cell line high-content drug screenings.** (**A**) Subclassification of the BLA-40 cell lines (*n* = 36) into the main known molecular BLCA subtype taxonomies: MDA, LUND, TCGA, and Consensus subtypes. The known subtypes converge into 3 main overlapping subtypes: basal, luminal, and mesenchymal-like. (**B**) Drug set enrichment analysis based on drug response data (percentage of cells alive) from the in-house HCS using 616 clinical inhibitors. Drugs were classified into 22 specific drug sets or classes. FDR = false discovery rate. (*p* < 0.033: *, *p* < 0.002: **, *p* < 0.001: ***) (**C**) HCS based on 3 representative cell lines per subtype. The heatmaps indicate the top 10 drugs with the highest differential inhibition of cell growth between subtypes, based on the z-score normalized cell counts.

**Figure 3 ijms-23-10605-f003:**
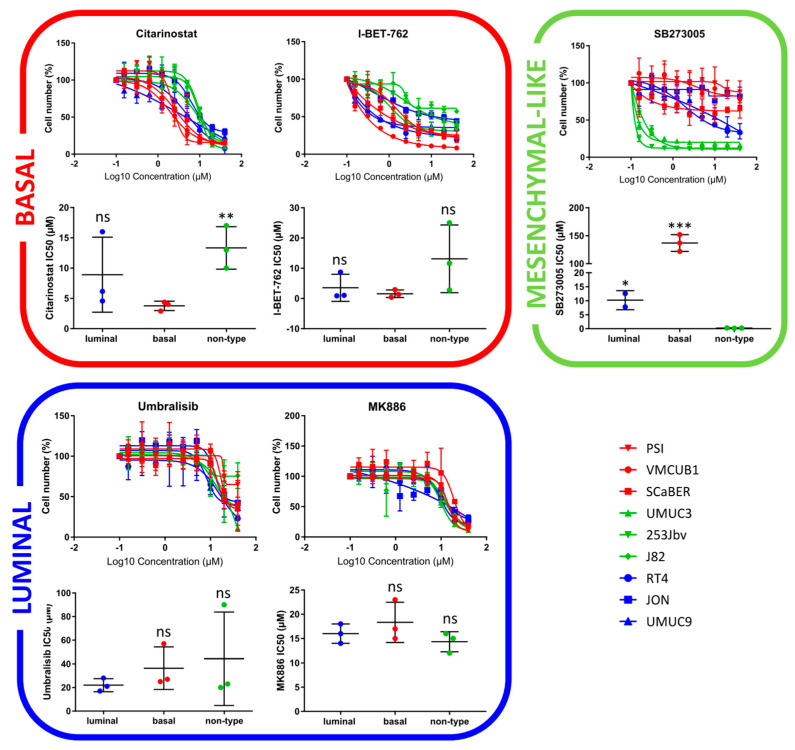
**Validation of lead hits via drug response curves in subtype-specific cell lines.** Basal: PSI, VMCUB1, SCaBER/Luminal: RT4, JON, UMUC9/Mesenchymal-like: UMUC3, 253Jbv, J82 (*p* < 0.033: *, *p* < 0.002: **, *p* < 0.001: ***).

**Figure 4 ijms-23-10605-f004:**
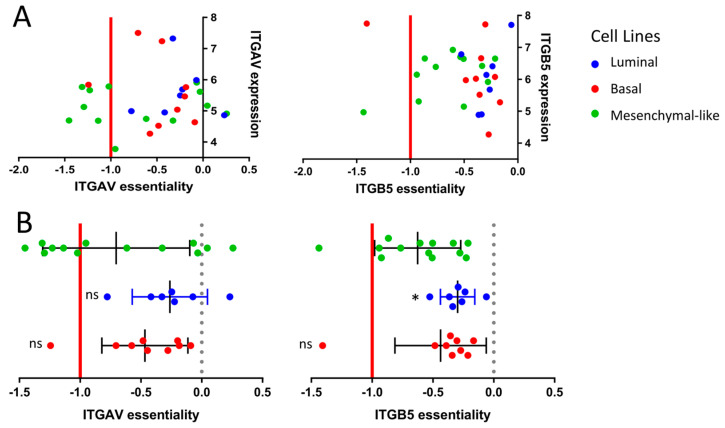
**Functional genomics based on the CCLE cell lines.** (**A**) Correlation between gene essentiality and gene expression based on the subtype-specific CCLE cell lines. The essentiality is calculated by the CERES dependency score based on the whole genome Crispr–Cas9 knock-out data from the Project Achilles. The x-axis displays the CERES dependency score: a score of 0 indicates that a gene is not essential for cell survival, and a score near −1 is comparable to the medians of all pan-essential genes. (**B**) Box-plot comparison of gene essentiality between the subtype-specific cell lines. (*p* < 0.033: *).

**Figure 5 ijms-23-10605-f005:**
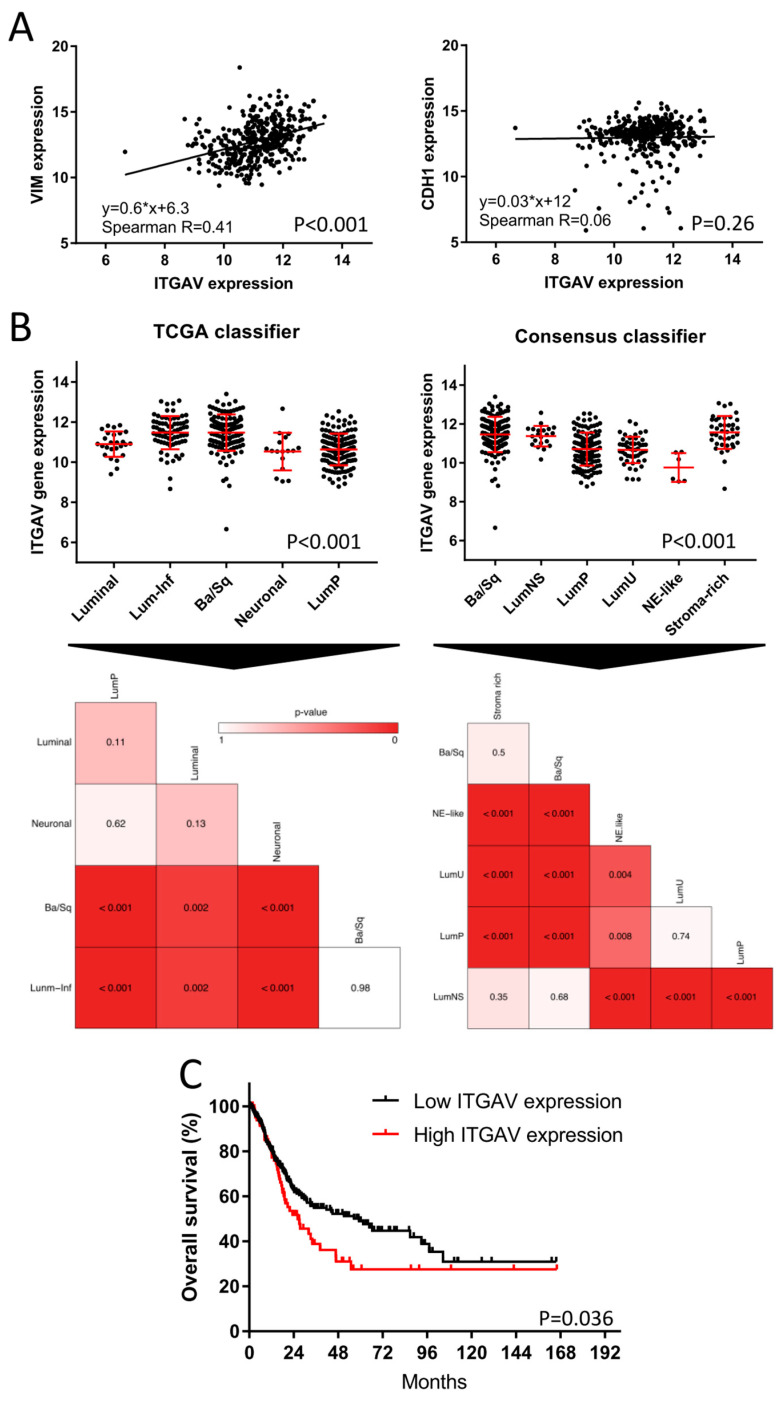
**Evaluation of *ITGAV* expression in bladder cancer patients of the TCGA cohort.** (**A**) BLCA TCGA Cohort (*n* = 386): Spearman correlation between *ITGAV* and vimentin (*VIM*) or *CDH1* gene expression as representative biomarkers for EMT. (**B**) Subtype-specific expression of *ITGAV* based on the TCGA and Consensus subtype taxonomies (*n* = 386). Heatmaps were used for the comparison of the statistical significance based on the differential gene expression of *ITGAV* between all subtypes. (**C**) Kaplan–Meier plot with log-rank comparison between overall survival of patients with high (*n* = 84) versus low *ITGAV* expression (*n* = 302).

**Figure 6 ijms-23-10605-f006:**
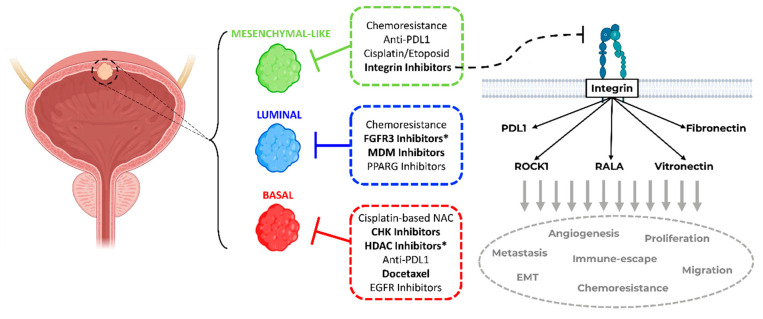
**Summary of personalized subtype-specific treatment options.** Proposed schema of subtype-stratified therapeutic approaches based on the current literature and findings of this study (in bold) as a framework for further prospective validation in clinical trials. (* Hits for bladder cancer jointly found by this study and in the literature. Bladder draft designed by BioRender).

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
