# Peer review of "High-Content Drug Discovery Targeting Molecular Bladder Cancer Subtypes"

_ijms, 2022, doi:10.3390/ijms231810605_

Round 1

Reviewer 1 Report

The authors presented an exciting strategy for identifying the novel subtype-specific treatment options and enhancing BLCA drug sensitivities. They suggest that a high-content screening (HCS) drug discovery approach could identify subtype-specific lead drugs, and gain insight into the molecular subtype biology. In particular, this study is meaningful in that it can indirectly improve its position for existing drugs by using transcriptomic data of various bladder cancer cell lines. In this study, it is regrettable that the author mainly planned the analysis for MIBC patients. This is because not only MIBC patients but also NMIBC patients have a subgroup in which cancer progression occurs well after surgery. Overall, the findings are clearly stated, and the discussion is written in a concise manner. I support the publication of this manuscript in IJMS, given that the authors address the below issues.

Words and figures issues

1.     Abstract: Molecular subtypes of muscle-invasive bladder cancer (BLCA) BLCA à MIBC

2.     Please review the legends in the figures again. In particular, we found that the expression styles of legends in Figures 1, 2, and 5 were different: (A), (B), c, and d.

3.     Please review the font size in the figures again. There are figures that are difficult to understand due to the large font size: Supplementary figures 3 and 4. It is difficult to understand the biological characteristics that the author wants to show through boxplots because of the large font size.

4.     Please review the order of all figures. From the reader’s point of view, supplementary figure 1 provides data to assist in selecting the 9 cell lines used for the author’s analysis. However, “Figure S1” is in the method section which is the end of the manuscript. We think the order of “Figure S1” should also be adjusted accordingly. In addition, “Figure 5C” is mentioned first before “Figure 5A”and “Figure 5B” in the text, but we think that it would be comfortable for the reader to read the manuscripts in the order of the figures. Thus it would be better to describe “Figure 5C” after “Figure 5A” and “Figure 5B” is mentioned.

5.     Please check the version of R studio used for analysis. The current version of R studio is 4.x.x (for desktop), but the version the author used is 1.x.x.

6.     Please check the use of italics font. Unnecessary use of italics was found throughout the manuscript. The expression “TCGA” did not require italics in the legends of Figure 5.

Analysis process issues

1.     The author classified the characterization of the bladder cancer cell line data of CCLE and the in-house using the previously known taxonomy of Lund, MDA, consensus, and TCGA. There are 5 subtypes in TCGA taxonomy (Luminal, Luminal papillary, Luminal infiltrated, Basal/squamous, and Neuronal). Among them, the “luminal infiltrated” is considered to be very important, but was excluded from this study (Figure 1 and 2). We think that the reason for the exclusion of the “luminal infiltrated” should be in the manuscript

2.     What is the reason for distinguishing the group with high and low ITGAV expression from the survival analysis of TCGA BLCA in Figure 5C? In general, we would consider dividing into two groups based on the median of ITGAV expression. In these cases, the distribution of high and low patients is close to 1:1. However, we are curious as to why the distribution of patients is 1 (high ITGAV expression, n=84): 4 (low ITGAV expression, n=302).

3.     Is there any reason the p-values are based on 0.033 throughout the analysis? Generally, it is divided into three categories: 0.05 (*), 0.01 (**), and 0.001 (***). Thus rationale on the author’s intention to present unusual classification criteria should be included.

Author Response

  1. Abstract: Molecular subtypes of muscle-invasive bladder cancer (BLCA) BLCA à MIBC

We changed BLCA into MIBC in the abstract.

  1. Please review the legends in the figures again. In particular, we found that the expression styles of legends in Figures 1, 2, and 5 were different: (A), (B), c, and d.

Expression styles were corrected.

  1. Please review the font sizein the figures again. There are figures that are difficult to understand due to the large font size: Supplementary figures 3 and 4. It is difficult to understand the biological characteristics that the author wants to show through boxplots because of the large font size.

Font size is optimized in fig. S3 and S4 now S2 and S3.

  1. Please review the orderof all figures. From the reader’s point of view, supplementary figure 1 provides data to assist in selecting the 9 cell lines used for the author’s analysis. However, “Figure S1” is in the method section which is the end of the manuscript. We think the order of “Figure S1” should also be adjusted accordingly. In addition, “Figure 5C” is mentioned first before “Figure 5A”and “Figure 5B” in the text, but we think that it would be comfortable for the reader to read the manuscripts in the order of the figures. Thus it would be better to describe “Figure 5C” after “Figure 5A” and “Figure 5B” is mentioned.

Thank you for the tip. Figure S1 became S9. And “Figure 5C” is now mentioned after “Figure 5A” and “Figure 5B”.

  1. Please check the version of R studio used for analysis. The current version of R studio is 4.x.x (for desktop), but the version the author used is 1.x.x.

This was indeed the wrong version. Thank you for the advice.

  1. Please check the use of italics font. Unnecessary use of italics was found throughout the manuscript. The expression “TCGA” did not require italics in the legends of Figure 5.

We addressed this point. In general, we used italics for gene names.

Analysis process issues

  1. The author classified the characterization of the bladder cancer cell line data of CCLE and the in-house using the previously known taxonomy of Lund, MDA, consensus, and TCGA. There are 5 subtypes in TCGA taxonomy (Luminal, Luminal papillary, Luminal infiltrated, Basal/squamous, and Neuronal). Among them, the “luminal infiltrated” is considered to be very important, but was excluded from this study (Figure 1 and 2). We think that the reason for the exclusion of the “luminal infiltrated” should be in the manuscript

Thank you for this input. We already explained this issue in line 99-102. We added a few comments to clarify that point.

  1. What is the reason for distinguishing the group with high and low ITGAV expression from the survival analysis of TCGA BLCA in Figure 5C? In general, we would consider dividing into two groups based on the median of ITGAV expression. In these cases, the distribution of high and low patients is close to 1:1. However, we are curious as to why the distribution of patients is 1 (high ITGAV expression, n=84): 4 (low ITGAV expression, n=302).

This is a legitimate question. In fact, the simple division of a cohort by the median value of a continuous variable involves a high risk of missing a clinical meaningful cutoff. The method used in this paper and others (see references 59 and 60) also intend to minimize the false positive rate of patients identified as high or low risk.

  1. Is there any reason the p-values are based on 0.033 throughout the analysis? Generally, it is divided into three categories: 0.05 (*), 0.01 (**), and 0.001 (***). Thus rationale on the author’s intention to present unusual classification criteria should be included.

The P-value asterisk criteria of the NEJM were adapted in graphpad Prism. Those criteria seem more stringent. We added a comment in line 443. The reviewer is right that other cutoffs are more usual.

Reviewer 2 Report

Dear Academic Editor:

I have reviewed the study carried out by Rinaldetti S, et al., entitled: “High-content drug discovery targeting molecular bladder cancer subtypes”. In summary, this study evaluates novel subtype-stratified therapeutic approaches based on high-content drug screening (HCS) for muscle-invasive urothelial carcinoma. Expression data from the CCLE transcriptome and BLA-40 cell lines were used for assignment of molecular subtypes in different cell lines categorized into three groups. To identify potential drugs specific to each molecular subgroup, two independent HCSs were used. 

This topic is of interest to readers of the journal. In general, the work is well written, with a logical sequence of the methodology to achieve the objectives of the study. The results are discussed with adequate and updated references. I would like to add a few comments that I think could be important from the point of view of daily clinical practice. Below I indicate my comments:

Comment 1:

One of the main characteristics of muscle invasive bladder carcinoma is its heterogeneity, both clinical, morphological, molecular, etc. When using specific cell lines in your study, how should the results be interpreted in the context of heterogeneity? Is this a possible limitation of the study?

Comment 2:

From a practical point of view, the molecular classification of muscle invasive urothelial carcinoma has several limitations (cost, response time, etc.). Despite not being an objective of the study, it is possible to identify biological markers that allow the classification of MIUC subtypes using these cell lines, such as by means of immunohistochemistry that allows its routine application in clinical practice.

Author Response

Comment 1:

One of the main characteristics of muscle invasive bladder carcinoma is its heterogeneity, both clinical, morphological, molecular, etc. When using specific cell lines in your study, how should the results be interpreted in the context of heterogeneity? Is this a possible limitation of the study?

This is a legitimate question. The molecular subtypes are able to identify risk groups and stratify MIBC survival. Thus, the molecular classification of the cell lines is clinically meaningful. The translational or clinical benefit becomes apparent especially upon our discovery of ITGAV as a predictive biomarker for MIBC survival.

The morphological heterogeneity is overlapping to a certain extend with the molecular subtypes. As shown in most classification studies, e.g. papillary tumors are mainly included in the luminal subtype and squamous carcinoma are exclusively classified as basal. We think that this drug screen study is the first of its kind with regard to the molecular heterogeneity of bladder cancer.

Comment 2:

From a practical point of view, the molecular classification of muscle invasive urothelial carcinoma has several limitations (cost, response time, etc.). Despite not being an objective of the study, it is possible to identify biological markers that allow the classification of MIUC subtypes using these cell lines, such as by means of immunohistochemistry that allows its routine application in clinical practice.

This is indeed a good point. The identification of molecular subtypes by means of immunohistochemistry is an ongoing matter of debate. To date, there are no comprehensive studies comparing the efficacy of subtype attribution via immunohistochemistry with subtype attribution via molecular genetics. The consensus classifier is already a single-sample classifier, but it still needs a reduction of its gene set in order to enable a translation into clinics (as done for breast cancer or diffuse large b-cell lymphoma). However, such reduced gene sets still need to be validated in large prospective clinical trials.